



# Benchmarking global hydrological and land surface models against GRACE in a medium-size tropical basin

Silvana Bolaños Chavarría[1,2], Micha Werner[3], and Juan Fernando Salazar[1]

[1]Grupo de Ingeniería y Gestión Ambiental (GIGA), Escuela Ambiental, Facultad de Ingeniería, Universidad de Antioquia, Medellín, Colombia
[2]Grupo de Investigación en Ingeniería Sostenible (GIS), Facultad de Ingeniería, Politécnico Colombiano Jaime Isaza Cadavid, Medellín, Colombia
[3]Water Science and Engineering Department, IHE Delft Institute for Water Education, Delft, the Netherlands

**Correspondence:** Silvana Bolaños Chavarría (sbolanos@elpoli.edu.co)

**Abstract.** The increasing reliance on global models to address climate and human stresses on hydrology and water resources underlines the necessity for assessing the reliability of these models. In river basins where availability of gauging information from terrestrial networks is poor, models are increasingly proving to be a powerful tool to support hydrological studies and water resources assessments. However, the lack of in-situ data hampers rigorous performance assessment, particularly in trop-

ical basins where discordance between global models is considerable. Remotely sensed data of the terrestrial water storage obtained from the GRACE satellite mission can, however, provide independent data against which the performance of such global models can be evaluated. Here we assess the reliability of six global hydrological models (GHM) and four land surface models (LSM) available at two resolutions. We compare Total Water Storage (TWS)'s modelled dynamics with TWS derived from GRACE data over the Magdalena-Cauca basin in Colombia, a medium-sized tropical basin with a comparatively well-

developed gauging network. We benchmark monthly TWS changes from each model against GRACE data for 2002–2014, evaluating monthly variability, seasonality, and long-term trends. TWS changes are evaluated at basin level, as well as for selected sub-basins with decreasing basin size. We find that the models poorly represent TWS for the monthly series, but they improve in representing seasonality and long-term trends. The high-resolution GHM W3RA model forced by the Multi-Source Weighted Ensemble Precipitation (MSWEP) is most consistent at providing the best performance at almost all basin scales,

with higher-resolution models generally outperforming lower-resolution counterparts. This is, however, not the case for all models. Results highlight the importance of basin scale in the representation of TWS by the models, as with decreasing basin area, we note a commensurate decrease in the model performance. A marked reduction in performance is found for basins smaller than $60,000$ km$^2$. Although uncertainties in the GRACE measurement increase for smaller catchments, the models are clearly challenged in representing the complex hydrological processes of this tropical basin, as well as human influences.

We conclude that GRACE provides a valuable dataset to benchmark global simulations of TWS change, in particular for those models with explicit representation of the internal dynamics of hydrological stocks, offering useful information for the continued improvement of large-scale hydrological and land-surface models of the global terrestrial water cycle, including in tropical basins.



## 1 Introduction

Total Water Storage (TWS) is a fundamental variable of the global hydrological cycle, representing the sum of all water storage components including water in rivers, lakes and reservoirs, wetlands, soil, and aquifers. TWS plays a key role in the Earth's water, energy, and biogeochemical cycles (Syed et al., 2008). It reflects the partitioning of precipitation into evaporation and runoff, and the partitioning of available energy of the surface between sensible and latent heat (Kleidon et al., 2014). Current interest in TWS is not only in the knowledge of the redistribution of the current body of water in the hydrological cycle, but is also essential for forecasting and gaining insight into the impacts of extreme events such as droughts and floods (Zhang, 2017). As an integrated measure of water availability, both surface and groundwater, the dynamic of TWS has significant implications for water resources management (Syed et al., 2008). For this reason, the monitoring of changes in TWS is critical for characterising water resources variability, and to improve the prediction of regional and global water cycles and interactions with the Earth's climate system (Famiglietti, 2004). Despite the acknowledged importance of this variable, integrated observations of TWS are largely unavailable, further confounded by the global decline in gauging networks (Hassan and Jin, 2016). Given the heterogeneity of the hydrology of river basins, comprehensive observation of TWS is very difficult due to insufficient in-situ observations of the diverse water storages and fluxes. Estimation of TWS and its change is commonly done through water balances and the use of models, and is often underestimated or extremely difficult to measure (Tang et al., 2010). Even many traditional analyses have assumed that at longer timescales and over large regions, change in TWS can be approximated as zero. This implies that in water balance studies it is common to ignore the long-term trends of TWS (Reager and Famiglietti, 2013).

At the global scale, there are two categories of hydrological models: Land Surface Models (LSM) and Global Hydrology Models (GHM). LSM focus on describing the vertical exchange of heat and water by solving the surface energy and water balance. These were originally developed by the atmospheric modelling community to simulate fluxes from the land to the atmosphere because of the crucial linkages between the land surface and climate. As the emphasis of LSM is on simulation of energy fluxes, these may not provide accurate simulation of water storage changes (Scanlon et al., 2018). GHM in contrast focus on solving the water balance equation and simulating catchment outlet streamflow. One of the primary differences between LSM and GHM is the more physical basis of LSM, including water and energy balances, compared to the more empirical water budget approaches included in most GHM. Additionally, GHM are increasingly modelling human interventions, including water use and water resources infrastructure (Veldkamp et al., 2018), which most LSM do not (Scanlon et al., 2018). Such differences may, however, gradually reduce as resolution and physical basis of GHM improve (Clark et al., 2017), with GHM and LSM converging into Earth-System models (Bierkens, 2015). The performance of these GHM and LSM models varies because of the different physical representation of land-surface processes, differences in model structure and physics, parameterization, and atmospheric forcing data (Zhang et al., 2017).



GHM and LSM are increasingly applied in assessing global water resources availability (Schellekens et al., 2017) and change (Sperna Weiland et al., 2012), as well as for supporting specific applications including flood forecasting (Gudmundsson et al., 2012; Alfieri et al., 2013), hydrological drought modelling (Pozzi et al., 2013), water scarcity (Veldkamp et al., 2017) and assessments of socio-economic dependencies on water resources in a changing climate (Viviroli et al., 2020; Wijngaard et al., 2018). The increasing use of these models to support water resources assessments and this increasing range of applications raises questions on the reliability of models, as reliable representation of short and long-term variations of key hydrological variables such as TWS is critical.

Benchmarking results from global models against streamflow observations shows these are able to provide adequate simulations despite a lack of calibration, and that performance improves with improved resolution (Sutanudjaja et al., 2018), but also shows that performance varies substantially across basins and observation sites, and is influenced by catchment size as well as basin elevation (Sutanudjaja et al., 2018). Performance of different LHM and GHM may moreover vary substantially depending on climatic and basins conditions, with particular differences in snow dominated as well as tropical, monsoonal climates (Schellekens et al., 2017). Most performance assessments focus on using observed discharge at basin outlets as the benchmark, the availability of which is limited in most basins with tropical and monsoonal climates (González-Zeas et al., 2019). Additionally, Clark et al. (2017) call for the scrutiny of not just the discharge at basin outlets in model inter-comparisons, but also of internal states and fluxes. Due to limitations in data, uncertainty, adequate parameterizations, and computational constraints on model analysis, assessing the models using only the outputs, in this case, discharge, could result in a good model performance but a poor representation of the internal states. Therefore, a complementary evaluation of models is necessary where TWS, as a fundamental state variable of the basin, must be considered.

Thanks to recent technological advances, remote sensing products provide an independent observation of TWS changes, such as is the case of the Gravity Recovery and Climate Experiment (GRACE) satellite data (Tapley et al., 2004). The GRACE set of satellites are able to detect changes in the Earth's gravity, which is influenced by large-scale water storage variations and transport on Earth. Since 2002, the GRACE satellites have monitored monthly changes in water mass as TWS increase or decrease resulting from climate variability and human interventions. Through monitoring the time variable gravity field these satellites provide a more direct estimate of global changes in TWS anomalies than that which can be obtained from models (Tapley et al., 2004; Scanlon et al., 2018). GRACE has been shown to provide the opportunity to observe water storage dynamics for large river basins and can contribute to better understanding of hydrology at larger temporal and spatial scales, such as are important for climate studies (Lettenmaier and Famiglietti, 2006). Although GRACE has important limitations due to its resolution (Chen et al., 2016), data from GRACE do provide a uniquely independent estimate of the distributed TWS in a river basin as water balance estimation based on observed data and models require gauging data (which are often deficient or insufficient) or data from reanalysis models, which are not direct observations. Advances in GRACE processing from traditional spherical harmonics to more recent mass concentration (mascon) solutions have increased the signal-to-noise ratio and reduced uncertainties (Scanlon et al., 2016). Since these data have become available, GRACE data have been used to validate global model outputs in several studies (see Scanlon et al., 2018; Schellekens et al., 2017, for an overview), though these comparisons have largely been at a global scale. Assimilation of TWS variability from GRACE has been shown to improve model results in





semi-arid basins (Tangdamrongsub et al., 2017; Schumacher et al., 2018), and how well selected models represent TWS fluxes has been assessed using GRACE data in tropical basins such as the Amazon (Pokhrel et al., 2013), though a comprehensive benchmarking of GHM and LSM in tropical basins has not previously been done.

In a previous study carried out in the Magdalena-Cauca river basin (MC basin henceforth) in Colombia (Bolaños et al., 2020), GRACE products were assessed through comparison with water balance-based estimates identifying the mascon product from the Jet Propulsion Laboratory as the best in representing TWS dynamics in the study area. They found the existence of long term trends related to the La Niña and El Niño extreme phases of ENSO. Also, GRACE data revealed that these trends are not uniform across the study area but the water depletion rate is more pronounced in the lower parts of the basin than it is in the upper basin. The long-term trends found in water storage help improve our understanding of the dynamics of water resources in response to climatic and anthropic variability. The importance of understanding these long-term trends lies in providing tools for adequate management of the water resource in terms of sustainability. Consequently, it is necessary that models adequately represent these dynamics in TWS as they have important effects on present and future water sustainability.

Through the comparison of averaged TWS from models with GRACE-based estimates for a medium size tropical basin, we identify both the potential as well as possible deficiencies of a set of 10 models comprising both GHM and LSM, and analyse the reasons for different model behaviours. We investigate the performance of this set of models for the MC basin in Colombia, which offers the unique opportunity as a tropical basin with a dominant monsoonal climate, but that also has an observation network that is reasonably extensive, despite the recent decline (Rodríguez et al., 2019). We benchmark these 10 models in the basin as a whole, as well as for selected sub-basins with progressively decreasing catchment sizes, using GRACE data from the Jet Propulsion Laboratory mass-concentration (JPL mascon) solution.

With the purpose to contribute to the understanding of the dynamic nature of TWS as well as to contribute to future LSM and GHM development and improvement, this study highlights the value of using water storage from GRACE, in addition to traditional water fluxes, as a benchmark in assessing global models. Assessing models using these recently available data of an important state variable such as TWS, can contribute to a better understanding of the hydrological cycle processes, with the improvement in the modeling and forecasting of hydrological variables in tropical basins, thus being conductive to better tools for decision-making around water management and sustainability. The relatively large set of LSM and GHM models considered in this study are obtained through the open access global Water Resources Reanalysis dataset developed in the eartH2Observe (E2O) research project, a collaborative project funded under the European Union's Seventh Framework Programme (EU–FP7) (Schellekens et al., 2017).

## 2 Data and Methods

### 2.1 Study area

The MC basin is the primary river basin system in Colombia. It occupies a major portion of the country in the tropical Andes, draining an area of $\sim 276{,}000$ km$^2$ which is about 25% of the total territory of Colombia (Fig. 1). It has its headwaters high up in the Colombian Andes at an elevation of about 3,700 m above sea level and runs for some 1,612 km before flowing into





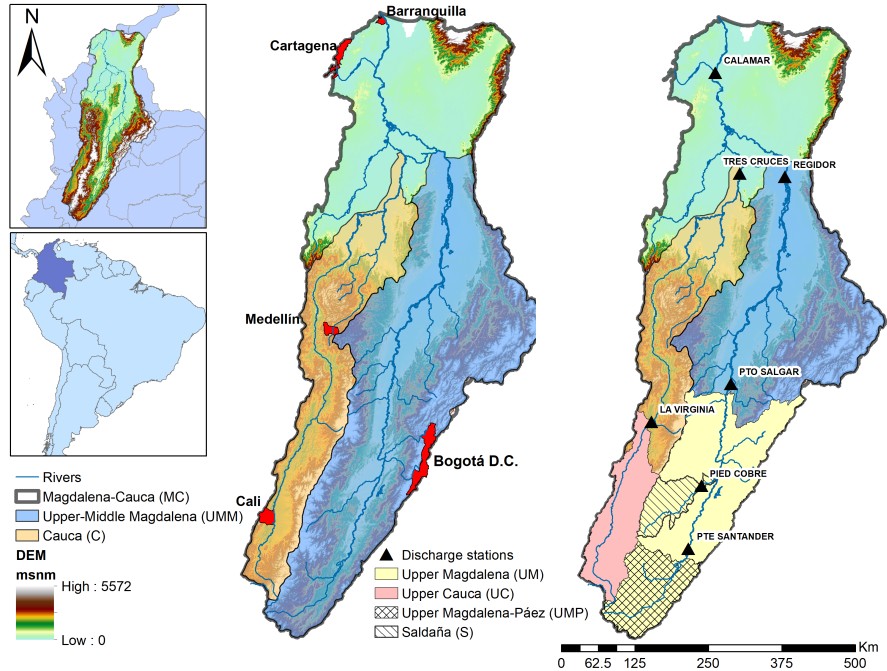

**Figure 1.** Location of the MC Basin in Colombia, as well as the sub-basins considered in this study. The triangles represent the locations of gauge stations measuring streamflow at the outlets of each (sub) basin.

the Western Caribbean, in the Atlantic Ocean (López López et al., 2018; Restrepo and Kjerfve, 2000). The main tributary of
the Magdalena River is the Cauca River, which flows along the western part of the basin and joins the main Magdalena River
in a wetland area called La Mojana, which is found in the Mompós Depression region. The mean annual river discharge at
the gauge station closest to the mouth (Calamar) is approximately 7,200 m$^3$ s$^{-1}$ with mean maximum discharges occurring in
November (10,200 m$^3$ s$^{-1}$), and minimum average flows in March (4,050 m$^3$ s$^{-1}$) (Camacho et al., 2008).

Due to the seasonal migration of the Intertropical Convergence Zone (ITCZ) around the equator, the climate in the basin
is on average bimodal, characterized by two wet periods (March–May and September–November) interspersed by two dry
periods. However, this is a basin with high variability, and we can observe that the lower MC tends to have a more unimodal
behavior (Urrea et al., 2019), with a continuous wet period from May to November (Fig. S1 in supplementary material).
The average annual precipitation in the basin is around 2,150 mm, while the annual average potential evapotranspiration is
estimated at 1,630 mm. The hydroclimatology of the basin is profoundly influenced by the El Niño - Southern Oscillation
(ENSO) phenomenon (Poveda and Mesa, 1996). Widespread flood events caused by the La Niña event of 2010–2011 affected
some four million Colombians and caused economic losses estimated at US 7.8 billion (Hoyos et al., 2013; Vargas et al., 2018),
while the severe droughts caused by the El Niño event of 2015–2016 had quite severe consequences, including water shortages





in more than 25% of the towns, the highest temperatures on record and numerous fires that impacted several regions in the country (Hoyos et al., 2017).

To analyse the effect of basin size on model performance and GRACE data, we subdivide the macro-basin into several sub-basins (Fig. 1). While the overall basin at Calamar station has an area of $\sim 276,000$ km$^2$, the Upper-Middle Magdalena (UMM) has an area of $\sim 140,754$ km$^2$; the Cauca (C) $\sim 60,657$ km$^2$; the Upper Magdalena (UM) $\sim 56,992$ km$^2$; and the Upper Cauca (UC) $\sim 17,930$ km$^2$. The smaller Upper Magdalena-Páez basin (UMP) has an area of $\sim 14,450$ km$^2$ and the Saldaña basin (S) $\sim 6,645$ km$^2$.

**2.1.1    GRACE data**

TWS anomalies from GRACE satellites are processed by three centers, the Center for Space Research at University of Texas (CSR), the Jet Propulsion Laboratory (JPL), and Geoforschungs Zentrum Potsdam (GFZ), using two different schemes, Spherical Harmonic (SH) and Mass Concentration (mascon) solutions. The similarities and differences between the SH and the mascon data are well explained by Scanlon et al. (2016) and Shamsudduha et al. (2017). Bolaños et al. (2020) evaluate the
different products of GRACE for the MC basin through the comparison with water balances based on observed data. They conclude that the best representation of TWS in the MC basin is the GRACE TWS product derived from JPL mascon. In this analysis we use the TWS anomalies data available from April 2002 through December 2014 from GRACE RL05 level-3 land JPL mascon solution gridded at 0.5° ($\sim 55$ km), based on an alternative processing approach which involves parameterizing the gravity field with regional mass concentration functions. This product has only recently become operational (Save et al.,
2016; Watkins et al., 2015; Wiese et al., 2016).

All reported data are anomalies relative to the 2004–2009 time-averaged baseline as presented in the original GRACE data. For consistency, all other data series used in this study are calculated as anomalies over their average values for the same period. The missing data due to battery management in GRACE were directly remedied by linear interpolation (Ouma et al., 2015; Xiao et al., 2015; Liesch and Ohmer, 2016; Shamsudduha et al., 2012). Variations in water mass or storage
are expressed as an equivalent water thickness (EWT; cm water). JPL mascon data were retrieved from the Tellus website https://grace.jpl.nasa.gov/data/get-data/jpl_global_mascons/.

**2.2    Earth2Observe global water resources reanalysis data**

In the analysis, input data were provided by the E2O project, which takes advantage of various global reanalyses and derived datasets to develop a global Water Resources Reanalysis (WRR) (Arduini et al., 2017; Dutra et al., 2015, 2017; Schellekens
et al., 2017). This dataset includes the outputs of 10 different Global Hydrological and Land Surface Models (GHM and LSM), which are available at two resolutions and time ranges, denoted WRR1 and WRR2. In WRR1, models were forced by the WATCH Forcing Data applied to the ERA Interim data (WFDEI) meteorological reanalysis dataset (Weedon et al., 2014) at a resolution of 0.5° ($\sim 55$ km at the equator) from 1979 to 2012. The WRR2 model runs were forced by the Multi Source Weighted Ensemble Precipitation (MSWEP) dataset (Beck et al., 2017) at a resolution of 0.25° ($\sim 27$ km at the equator)
from 1980 to 2014. Model algorithms were also improved between WRR1 and WRR2, such as by a better representation





of hydrological processes, incorporation of anthropogenic influence, and by integrating earth observation data (Gründemann et al., 2018). Arduini et al. (2017), Dutra et al. (2015), Dutra et al. (2017), and Schellekens et al. (2017) provide a detailed description of the two datasets and the model improvements. Table 1 provides an overview of models considered in this study, as well the main changes in the models between WRR1 and WRR2 for those models that have been run using both forcing

datasets and at both resolutions. The performance of several of these models has been compared over the MC basin, finding that key water resources management indicators derived using these models compare well against those derived using in-situ data (Rodríguez et al., 2019).

In order to compare the different models and the WRR with TWS obtained from GRACE JPL mascon, data for the period from 2002 to 2012 were used in this study as a common period for WRR1 and GRACE, and 2002 to 2014 for WRR2 and

GRACE. Data were downloaded from the E2O Water Cycle Integrator portal for the required period and for the required spatial domain (https://wci.earth2observe.eu/, last access: 20 November 2018).

### 2.3 Assessment of model performance

Monthly change in TWS can be calculated as the result of water balance estimates as presented in Equation 1, where $S$ is the terrestrial water storage, $P$ and $E$ are the basin-wide totals of precipitation and actual evapotranspiration and $R$ represents total

basin outflow, or the net surface and groundwater outflow. Changes in TWS can also be computed as the sum of the monthly changes in component storages, presented in Equation 2; where $\Delta GWS$ is the change in groundwater storage, $\Delta SMS$ the change in soil moisture storage, and $\Delta SWS$ the change in surface water storage. $\Delta CWS$ is the change in canopy water storage and $\Delta SWE$ represents the change in snow water equivalent, which is not considered in this study because the area under snow influence represents less than 0.1% of the total area of the basin.

$$\Delta TWS = \frac{dS}{dt} = P - E - R, \tag{1}$$

$$\Delta TWS = \Delta GWS + \Delta SMS + \Delta SWS + \Delta CWS + \Delta SWE. \tag{2}$$

Not all of the models contained in the E2O dataset explicitly represent groundwater storage. For those models that explicitly represent groundwater storage as well as surface and soil moisture components we apply both equations for evaluating simulated TWS. For models that include only surface water and soil moisture components we cannot apply the second equation

(Eq. 2), which allows a more representative evaluation of TWS as GWS is the largest water component of TWS on land.

Table 2 shows the variables for each model available in the E2O Water Cycle Integrator portal. The table also provides an overview of which of the above equations to derive simulated TWS changes are applied in the two available model resolutions (WRR1 and WRR2). The symbols and colour in the last four columns identify the datasets and equations considered given their availability, and are used throughout the paper.

In order to understand the variation between TWS change from GRACE JPL mascon and simulated TWS from models at the basin scale, both model and GRACE time series were disaggregated using the Seasonal Trend decomposition by Loess



**Table 1.** Overview of models and main changes from WRR1 to WRR2. Note that not all models included in WRR1 were run for WRR2, in which case no changes are noted in the table.

| Model | Provider Organisation | Model type | Model changes in WRR2 | Reference |
|---|---|---|---|---|
| HBV–SIMREG | Joint Research Centre (JRC) | GHM | n/a | Lindström et al. (1997) |
| LISFLOOD | Joint Research Centre (JRC) | GHM | Increased number of soil layers, groundwater abstraction. | Van Der Knijff et al. (2010) |
| PCR–GLOBWB | Universiteit Utrecht (UU) | GHM | Water use included. Improvements to river routing reservoir schemes and water withdrawal and consumption. | Van Beek et al. (2011) |
| SWBM | Eidgenössische Technische Hochschule (ETH) | GHM | n/a | Orth and Seneviratne (2013) |
| W3RA | Eidgenössische Technische Hochschule (ETH) | GHM | Modified soil and groundwater hydrology equations, improved parameter estimates, dynamic data assimilation, evaporation of water not derived from rainfall. | Van Dijk et al. (2014) |
| WaterGAP3 | Universitat Kassel | GHM | Assimilation of soil water estimates, reservoir management. | Flörke et al. (2013) |
| HTESSEL | European Centre for Medium-Range Weather Forecasts (ECMWF) | LSM | Multi-layer snow scheme, increased number of soil layers. | Balsamo et al. (2009) |
| JULES | Centre for Ecology and Hydrology (CEH) | LSM | Rainfal-runoff processes, inclusion of a terrain slope dependency in the saturation excess runoff scheme. | Best et al. (2011) Clark et al. (2011) |
| ORCHIDEE | Centre National de la Recherche Scientifique (CNRS) | LSM | Revision of the ancillary data, surface roughness, snow scheme, soil freezing and routing. | Krinner et al. (2005) |
| SURFEX-Trip | Meteo France | LSM | Improvements in ground water, flood plains, land use, plant growth, surface energy and snow. | Decharme et al. (2010) |

Source: Schellekens et al. (2017); Dutra et al. (2017).

n/a: not applicable.





**Table 2.** Components used in TWS change estimation for each model.

| Model | Evaporation* | Runoff* | Variables WRR1 | Variables WRR2 | WRR1 Eq.1 | WRR1 Eq.2 | WRR2 Eq.1 | WRR2 Eq.2 |
|---|---|---|---|---|---|---|---|---|
| HBV-SIMREG | Penman 1948 | Beta function | P, ET, R, SMS, GWS, CWS | | ▲ | △ | | |
| LISFLOOD | Penman–Monteith | Saturation and infiltration excess | P, ET, R, SMS, GWS | | ▼ | ▽ | | |
| PCR-GLOBWB | Hamon (tier 1) or imposed as forcing | Saturation and infiltration excess | P, ET, R, SMS, GWS, CWS, SWS | P, ET, R, SMS, GWS, SWS | ● | ○ | ● | ○ |
| SWBM | Inferred from net radiation | Inferred from precipitation and soil moisture | P, ET, R, SMS | | * | | | |
| W3RA | Penman–Monteith | Saturation and infiltration excess | P, ET, R, SMS, GWS | ET, R, SMS, GWS | ■ | □ | | □ |
| WaterGAP3 | Priestley–Taylor | Beta function | P, ET, R, CWS, SWS | P, ET, R, CWS, SWS | + | | + | |
| HTESSEL | Penman–Monteith | Saturation excess | P, ET, R, SMS, CWS | P, ET, R, SMS | ▲ | | △ | |
| JULES | Penman–Monteith | Saturation and infiltration excess | P, ET, R, SMS, CWS | P, ET, R | ■ | | □ | |
| ORCHIDEE | Bulk PET | Green-Ampt infiltration | P, ET, R, SMS, SWS | | X | | | |
| SURFEX-Trip | Penman–Monteith | Saturation and infiltration excess | P, ET, R, SMS, GWS,CWS,SWS | P, ET, R, SMS, CWS,SWS | ● | ○ | ● | |

* Schellekens et al. (2017)





(STL) proposed by Cleveland et al. (1990) to estimate the relative magnitudes of water storage variance of different time series components (Eq. 3):

$$\Delta TWS = \Delta TWS_{long-term} + \Delta TWS_{seasonal} + \Delta TWS_{residuals}. \tag{3}$$

Hydrological performance of the monthly simulated TWS changes from all models was assessed using commonly used model evaluation statistics. We consider Pearson's correlation coefficient (r), Root Mean Square Error (RMSE), Ratio of RMSE to the standard deviation of the observations (RSR), and the Kling-Gupta efficiency (KGE; Gupta et al., 2009). Pearson's correlation coefficient (r) provides an indication of the linear relationship between simulated TWS and the benchmark TWS derived from GRACE. RMSE indicates how close model predicted values are to observed data, estimating the square root of
the variance of the residuals. Lower values of RMSE indicate better fit. RSR standardizes RMSE using the standard deviation of the observations, and is calculated as the ratio of the RMSE and standard deviation of the observed data. RSR varies from the optimal value of 0, which indicates zero RMSE or residual variation and therefore perfect model simulation, to an infinitely large positive value. The lower RSR, the lower the RMSE, and the better model performance (Moriasi et al., 2007). Finally, the KGE index facilitates analysis of the relative importance of different components in the context of hydrological modelling.
In the computation of this index, there are three main components involved: the Pearson's correlation, the ratio between the standard deviation of the simulated values and the standard deviation of the observed ones, and the ratio between the mean of the simulated values and the mean of the observed ones. KGE ranges between -∞ and 1, where 1 indicates a perfect representation of TWS.

## 3   Results

### 3.1   TWS evaluation for the whole MC basin

In order to evaluate how well the models represent TWS for the whole basin, we apply Eq. 3 to analyse the monthly series, seasonality and long-term trends of TWS for each of the model datasets against the GRACE data. Results are shown in the Taylor diagrams in Figure 2, which provide a $2D$ graphical representation of three statistics to indicate how well the simulated pattern matches that of the observations. Similarity is quantified in terms of the correlation(s), the ratio(s) of the normalized
RMSE differences between simulated and observed, and the amplitude of their variations represented by the ratio of the standard deviations of simulated and observed (Taylor, 2001). Best performances are for values of correlation close to 1, with a low RMSE, and a ratio of standard deviations close to 1. Figure 2 shows Taylor Diagrams for the complete monthly series (a), for the seasonality (b), and long-term trends (c). In all cases the corresponding constituents of the GRACE JPL mascon dataset are used as reference. In this figure, as in all further figures, a blue colour is used to represent the GHM in WRR1 and
a green colour is used for the GHM in WRR2. The LSM in WRR1 are represented with a red colour, while LSM in WRR2 are shown in yellow (see also Table 2). Model runs derived from WRR1 are denoted as "R1" for brevity, while those derived from WRR2 are denoted as "R2", as well as models denoted "Eq1" or "Eq2" refer to the Equation used to estimate TWS.

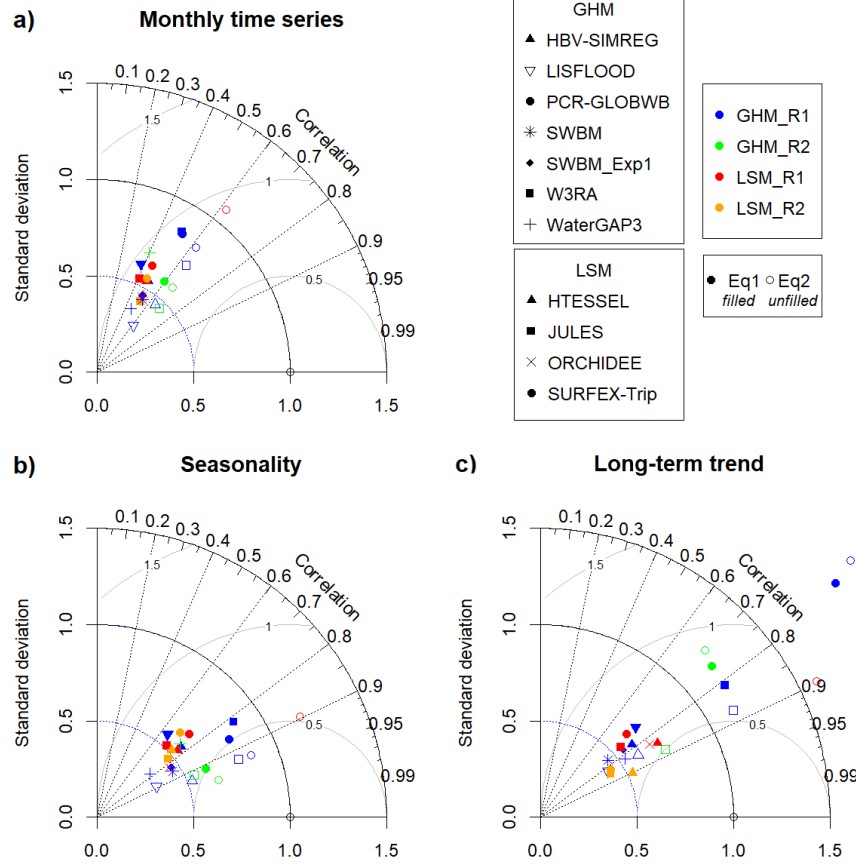

**Figure 2.** Taylor diagrams between each model output and GRACE data for the Magdalena river basin for a) monthly time series, b) seasonality, c) long-term trends. R1 indicates the models from the first reanalysis WRR1, and R2 the second reanalysis WRR2.

From Figure 2 it is not possible to distinguish clearly which of the models in total 23 data sets from GHM WRR1, GHM WRR2, LSM WRR1 and LSM WRR2 better represents TWS when compared to GRACE. It is noteworthy that correlations are better for almost all models when considering seasonality and long-term trends. For the monthly series, the correlations of the models range between 0.36 and 0.68, with the highest correlation corresponding to the W3RA R2Eq2 and the lowest to LISFLOOD R1Eq1. However, for almost all models (except Surfex-Trip R1Eq2) smaller standard deviations are found than those observed by GRACE. For the representation of seasonality, we observe that correlations increase in all models, with the highest correlation found for PCR-GLOBWB R2Eq2 ($r = 0.96$) and the lowest for LISFLOOD R1Eq1 model ($r = 0.65$). We also observe that the models with good correlation and whose standard deviations are closer to those observed are GHMs, particularly PCR-GLOBWB R1Eq1, PCR-GLOBWB R1Eq2, W3ERA R1Eq1 and W3RA R1Eq2 models. For long-





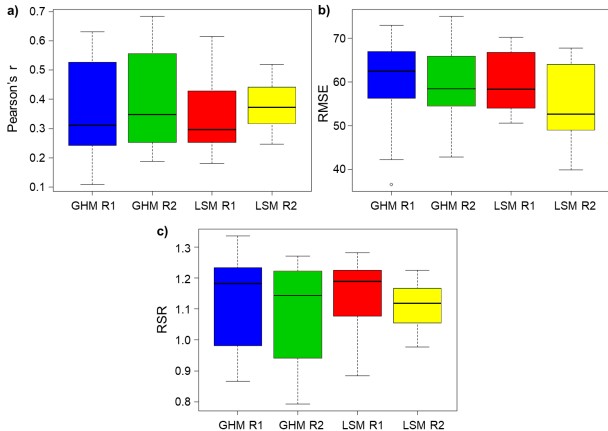

**Figure 3.** Distribution of model performance for the three metrics considered, grouped by model type (GHM or LSM) and forcing/resolution (WRR1 and WRR2). Three performance metrics are shown; (a) Pearson's r, (b) RMSE, and (c) RSR.

term trends, an increase in correlations is also observed. The highest correlation is now found for HTESSEL R2Eq1 model (r = 0.91) and the lowest for WaterGAP3 R2Eq1 (r = 0.23). Standard deviations for most models are lower than those observed, indicating too low a variability. WaterGAP3 R2Eq1, PCR-GLOBWB R1Eq1 and Eq2, and SurfexTrip R1Eq2 differ in that
these show very high standard deviations compared to those observed in GRACE TWS.

### 3.2 TWS monthly values evaluation

To provide an overview of the range in performance metrics comparing TWS at the sub-basin scale to GRACE derived TWS, Figure 3, groups the correlation, RMSE, and RSR by model type (GHM and LSM) and reanalysis (WRR1 and WRR2). We observe an improvement in the performance of the WRR2 models over the WRR1 models, with an increase in correlation
values and decrease for both the RMSE and RSR.

This is further explored in Figure 4, showing the relationship between TWS of the sub-basins area of the MC basin and the error statistics for the models in WRR1 and WRR2. To allow comparison, error statistics are normalised and standardized. These results clearly demonstrate the detriment in model performance as basin size decreases. This is best observed in the KGE statistic (Fig. 4a). It is evident that the models are generally able to better capture the hydrology for the main basins in both
WRR1 and WRR2. For the UM sub-basin (56,992 km$^2$) and the smaller sub-basins there is a marked decrease in performance, although the only slightly larger Cauca basin (60,657 km$^2$) shows much better performance. This provides an indication of the basin size at which the models are capable of capturing TWS and also illustrates the difference in forcing, resolution and the model's improvements made in WRR2 over WRR1. For WRR1 the best performance for the MC and UM basins is found for HBV-SIMREG R1Eq2 model; the W3RA R1Eq2 model performs reasonably well for the UMM and Cauca basins; and the
SWBMExp1 R1Eq1 model has the best performance for UC, UMP and Saldaña basins. All three of these models are GHM. For WRR2, the best performance in almost all basins is found for the W3RA R2Eq2 model, though for the smaller basins



(UMP and Saldaña) it is found to be the JULES R2Eq1 model. The first of these is a GHM and the second an LSM. The models with the lowest KGE values are LISFLOOD R1Eq1 for MC, UMM and UM basins, WaterGAP3 R2Eq1 for Cauca, and PCR-GLOBWB R1Eq2 for the last three basins. For models that have been run both for WRR1 and WRR2, it is not

consistently found that the WRR2 runs have improved performance for all basins. For instance, PCR-GLOBWB WRR1 is better than WRR2 for only the Cauca basin, while for HTESSEL WRR1 is better than WRR2 for the Cauca and UC basins. Also, WaterGAP3 presents better performance for WRR1 in all basins, while Surfex-Trip has highest values for WRR1 than for WRR2 for all basins except the two smaller basins. For W3RA the best performance is consistently found for WRR2 across all basins.

In Figure 4b we display the Pearson's correlation coefficient $r$ for each model. For both WRR1 and WRR2, the best performances are consistent with the KGE index. With some exceptions, the WRR2 models show improved performance over WRR1. The model with the highest correlation in all basins is W3RA R2Eq2. Models with the lowest RMSE alternate between W3RA R2Eq2, WaterGAP3 R1Eq1 and JULES R2Eq1. WaterGAP3 is the opposite case, and SURFEX-Trip has an improvement in WRR2 over WRR1 only in the UMP and Saldaña basins. The RMSE in Figure 5c shows consistency with the other statistics.

The lowest values correspond again to W3RA R2Eq2 and JULES R2Eq1 showing the best performance.

    Even though some models perform relatively well, the overall performance of the models is in general, poor. Average KGE remains negative for all models and sub-basins, and the average Pearson's $r$ value does not exceed $0.5$ in most cases. We highlight the decrease in the performance of all models for the smaller basins. The shift between the Cauca and UM basins, with areas of $\sim 60{,}657$ km$^2$ and $\sim 56{,}992$ km$^2$ respectively is interesting, and although the specific catchment conditions in

these two basins may lead to what appears a step change, the reduction of performance of the models for the smaller basins is clear. It is worth noting that GRACE resolution will begin to have more limitations in these smaller basins.

    Figures 5 and 6 show the spatial relationship between the TWS monthly values of GRACE and the models with both reanalysis WRR1 and WRR2, and the models with only WRR1 respectively. In this figure we rescale the GRACE TWS and the models from 0.5° to 0.25° using bilinear interpolation to interpolate from one rectilinear grid to another, to be consistent with

the finest resolution WRR2 models. The W3RA Eq2 WRR2 is the model with the highest correlation values. The maximum values presented in the maps are close to 0.8 and are located mainly towards the North of the macro-basin, with the minimum values towards the South. This is coherent with Figure 4, in which we observe a reduction in performance of all models upstream of the UM basin. This suggests that is not only basin size that is relevant to the performance of each model. The prevailing pattern could suggest it is related to hydrological process, or more likely to the presence of storages in these areas

that the models are not properly simulating. In Figure 5 it is also clear that there is an improvement in the spatial correlations of models when forced with WRR2 rather than WRR1, except for WaterGAP3.

## 3.3   TWS seasonality and evaluation of long-term trends

To assess the seasonal signals and long-term trends of the models against GRACE TWS we show in Figure 7 the Pearson's r coefficient and RMSE statistic for all sub-basins. We observe a similar pattern as in the Taylor Diagrams presented in Figure

2 for the macro-basin, but now highlighting model performance with decreasing basin size. We observe the same shift in the





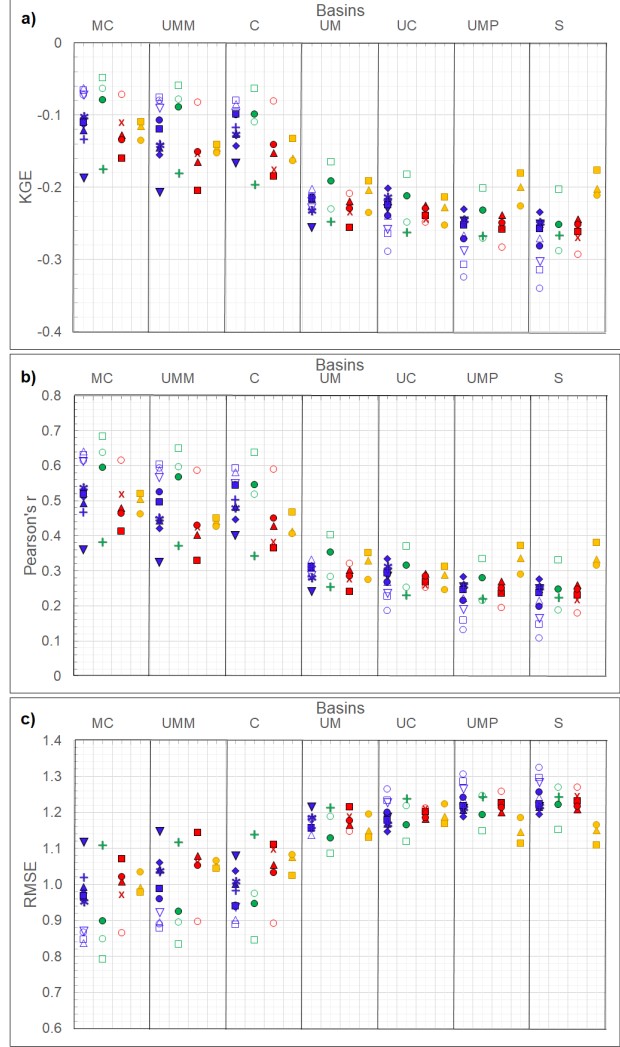

**Figure 4.** Performance statistics between GRACE and the GHM (blue and green points for WRR1 and WRR2 respectively) and LSM (red and yellow points for WRR1 and WRR2) at different scales. In a) the KGE index, b) Pearson's r, and c) the RMSE. The basins are ordered by size (largest to the left), and the data is standardized and normalized.

performance of models for basins of the size of the UM basin and smaller both for seasonality (Fig. 7a,b) and long-term trend (Fig. 7c,d). The Pearson's r values for both is better than in the monthly values analysis (Fig. 7a,c). For seasonality, the highest r coefficient is found for PCR-GLOBWB R2Eq2 for the MC and UMM basins, and for W3RA R2Eq2 for the remaining sub-basins, which coincides with the lowest RMSE in Figure 7b. For long-term trends, the highest r coefficient is presented in HTESSEL R2Eq1 for MC, JULES R2Eq1 for UMP, and W3RA R2Eq2 for remaining sub-basins, which coincides with the lowest RMSE in Figure 7d. WaterGAP3 R2Eq1 presents the lowest performance for all sub-basins in the long-term trends.



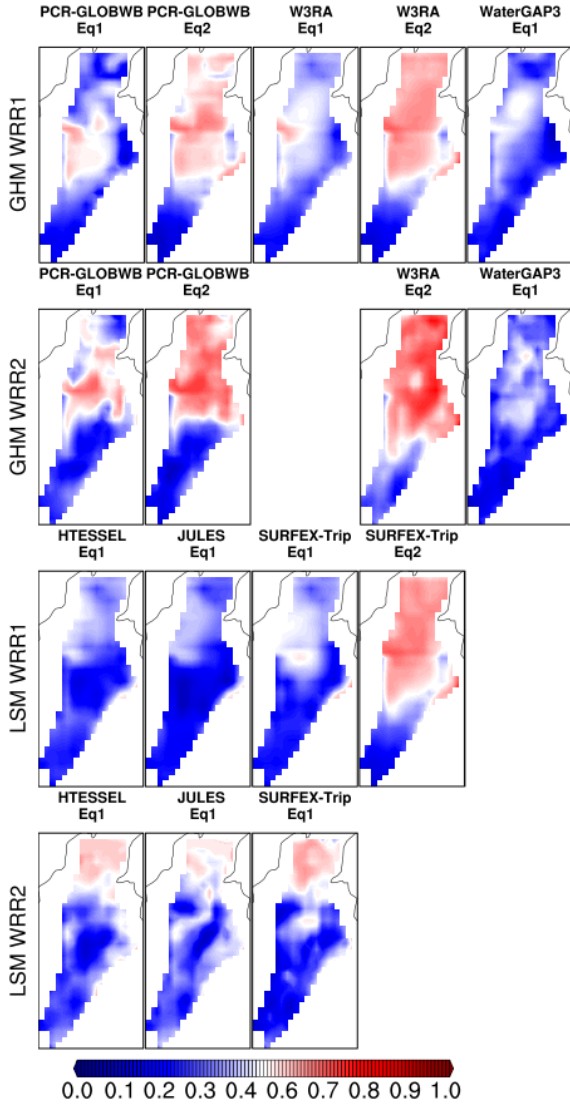

**Figure 5.** GRACE JPL vs Models correlation maps for whose that are available both in WRR1 and WRR2 for Eq1 and Eq2.

The seasonality found for the GRACE data, as well as for each group of models and for each sub-basin is presented in Figure 8. In general, we observe good agreement between TWS from the models and GRACE for the main basins. The bimodal behaviour in all models and in the GRACE data is consistently represented, as a consequence of the dominant bimodality of precipitation in the whole MC basin (Poveda, 2004). However, for the UM basin and the smaller basins, the models tend to overestimate the second peak in the SON (September–October–November) season. The maximum peak in GRACE for all sub-basins occurs in the month of May, while in the models it varies.





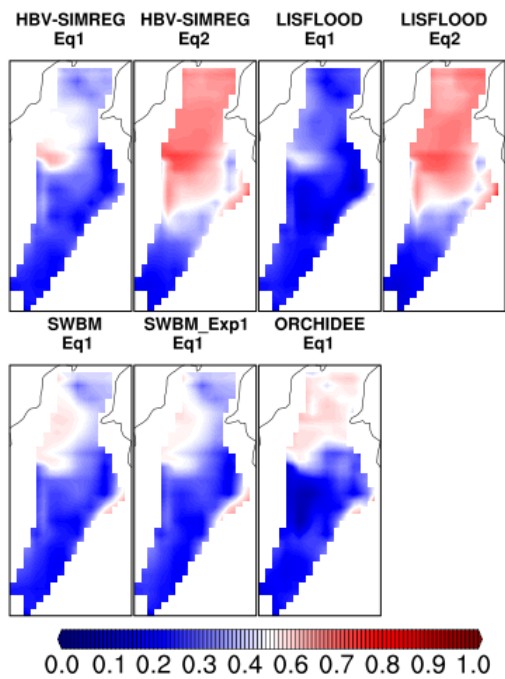

**Figure 6.** GRACE JPL vs Models correlation maps for whose that are available only in WRR1 for Eq1 and Eq2.

Figure 9 compares the seasonal maps of TWS estimated for GRACE and the models with the highest correlations following the Figures 5 and 6 and models that perform similar. The seasonal maps of TWS for the other models are shown in the supplementary material (Figs. S2 and S3). Here, we observe that the amplitudes for most models is smaller than for the GRACE data. This implies that the models tend to underestimate seasonal variation. For DJF (dry season) the models tend to be consistent with GRACE, with the lowest biases found in the North. In general, for the basin the values of TWS are negative or near zero. On the other hand, MAM (the first wet season), shows the opposite case with positive values throughout the macro-basin, with some exceptions towards the North like the W3RA R2Eq2. In JJA (dry period) we again observe consistency. There is a transition between the two rain periods, with positive bias in the North and negative bias in the South of the basin. In SON the models differ spatially with GRACE. The highest values of TWS from GRACE are found in the North of the basin in SON. This part of the basin is dominated by La Mojana wetlands. On the contrary, the models mostly present biases in the area of La Mojana close to 0, and higher TWS values towards the South of the basin. These higher values in the South correspond with the peaks observed in Figure 8. This behaviour is likely due to the poor representation of the wetlands in the models. The buffering capacity in the models is poorly represented, which means that these dry out too much in the drier DJF period and cannot represent the wet period in SON in La Mojana area, which on can be observed by GRACE. For the second dry season, the fact that the climate becomes more unimodal to the North (see Fig. S1) may also contribute. Both drying and wetting may also be overestimated by the models as observed in the South, specially, during JJA and SON, and SURFEX-Trip R1Eq2 during DJF.

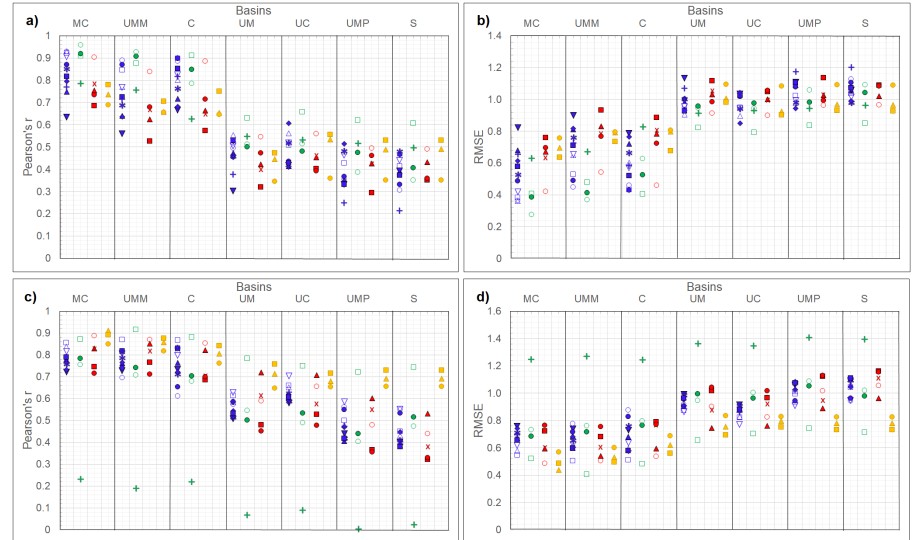

**Figure 7.** Performance statistics for the seasonality (a–b) and long-term trends (c–d) of GRACE and the GHM (blue and green points for WRR1 and WRR2 respectively) and LSM (red and yellow points for WRR1 and WRR2) at different scales. In a) and c) the Pearson's r coefficient and in b) and d) the RMSE.

Finally, we explore the agreement between the models and GRACE for the long-term component. Figure 10 shows the series
for GRACE JPL (black line) as well as for each model group GHM WRR1, GHM WRR2, LSM WRR1 and LSM WRR2 (blue, green, red and yellow lines respectively). We present the graphs for MC, Cauca, UM, and Saldaña basins, as the other sub-basins are similar to these and are shown in the supplementary material (Fig. S4). The MC and Saldaña basins are the largest and smallest basins respectively, while the change in model performance occurs between Cauca and UM basins. Large discrepancies between models and GRACE can be seen, with these increasing as basin size decreases. We observe that WaterGAP3 R2Eq1,
PCR-GLOBWB R1Eq1 and Eq2, and SURFEX-Trip R1Eq2 (the first two GHM and the last LSM), overestimate both highs and low peaks. By contrast, LISFLOOD R1Eq1 and SWBM R1Eq1 outputs are relatively flat, underestimating both the highs and the lows. However, all models are able to capture the increase in TWS during the 2010-2011 ENSO event (La Niña), likely due to the (common) precipitation forcing. Better performance is shown for the LSM than for GHM in general, although the results for W3RA R2Eq2 are closest to observations.

## 4 Discussion

### 4.1 Evaluation of the performance of the models

To summarize the results of the performance of the models with respect to GRACE, we present the performance metrics for each model and WRR in Figure 11. Higher score values (blue boxes) correspond to a better performance in representing





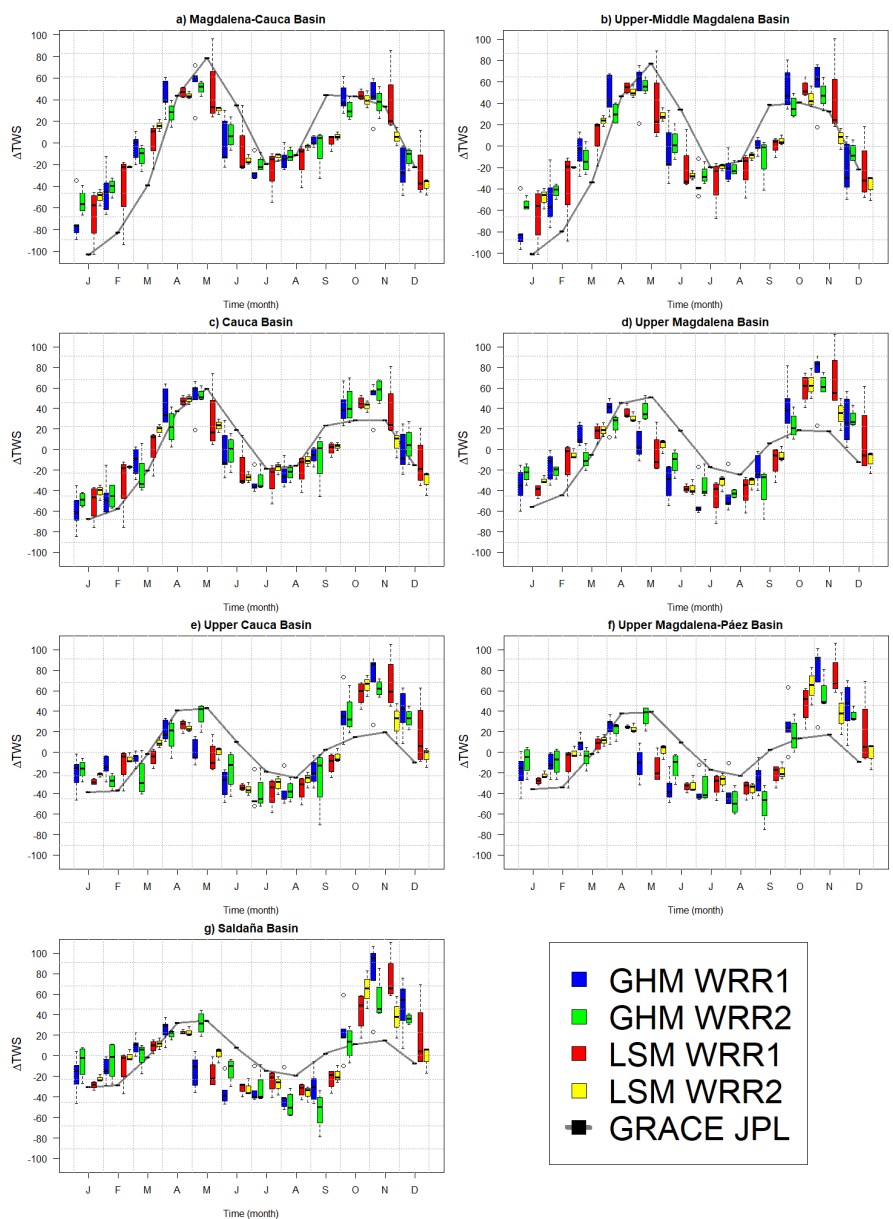

**Figure 8.** Comparative boxplots between the climatology of GHM and LSM, and GRACE JPL for each subbasin.

TWS, while lower score values (red boxes) indicate a poor representation for monthly time series, seasonality and long-term

trends. This shows that the WRR2 models tend to exhibit better performance than the lower resolution WRR1. This is coherent

with results of Gründemann et al. (2018), who assessed simulated discharges from seven of the ten models studied here, as

well as the ensemble mean of those models, focusing on the occurrence of floods in the Limpopo Basin in Southern Africa.



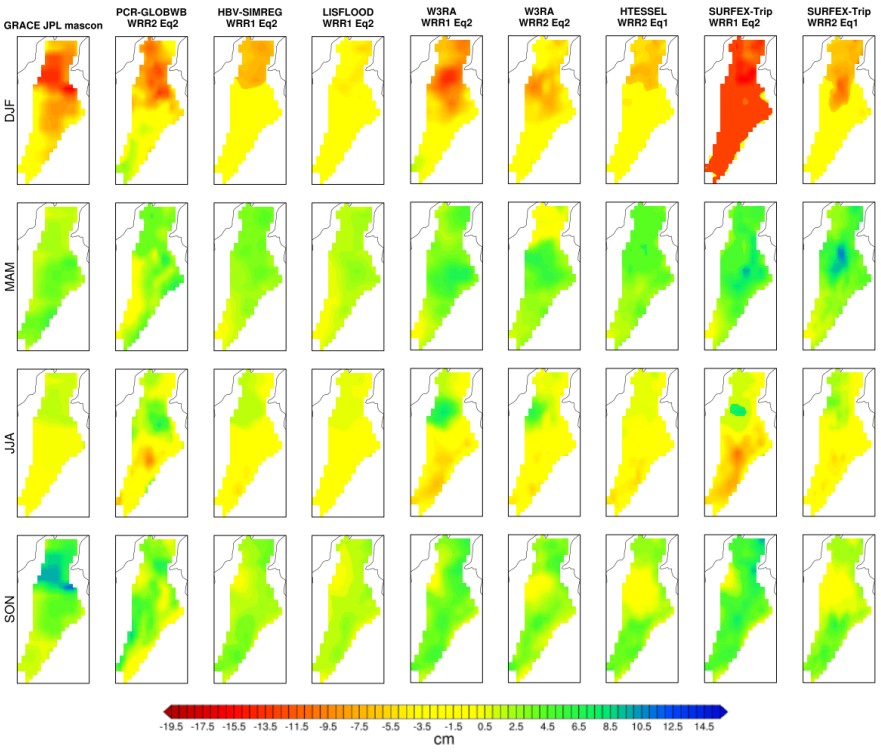

**Figure 9.** Seasonal maps for GRACE JPL, GHM, and LSM.

The exception of this improved performance is found for WaterGAP3, which shows much poorer performance for WRR2, in particular in the representation of long term trends.

Since the models have the same input/forcing for each WRR, differences in simulation results can only be attributed to the model structure and internal model dynamics. A number of factors could contribute to the notably poor performance of WaterGAP3 in WRR2 for the study area. These include modifications made to the simulation of hydrological processes and how reservoir management is represented. Dutra et al. (2017), however, note that the modification made for WRR2 primarily affect the water stored in the reservoirs and not the surface runoff, evapotranspiration, or other surface fluxes. While there

are a large number of reservoirs in the basin, these are primarily used for hydropower generation and to a lesser extent to serve irrigation, flood control or other purposes. As a result, the degree of regulation of these reservoirs is found to be quite low (Angarita et al., 2018). The reservoir release scheme in the WaterGAP model follows a more generic concept independent of the primary purpose of the reservoir (Müller Schmied et al., 2021), and thus may misrepresent the dynamics of the reservoir storage in this basin. WaterGAP3 was also found to perform poorly with respect to other global hydrological models in a snowmelt-

driven catchment in northern Canada that also includes extensive regulation for hydropower production (Casson et al., 2018). In contrast, however, for the Limpopo River basin in Southern Africa, the WaterGAP3 model in WRR2 demonstrated the best performance in simulating flood events among the same models considered here (Gründemann et al., 2018). This is also an



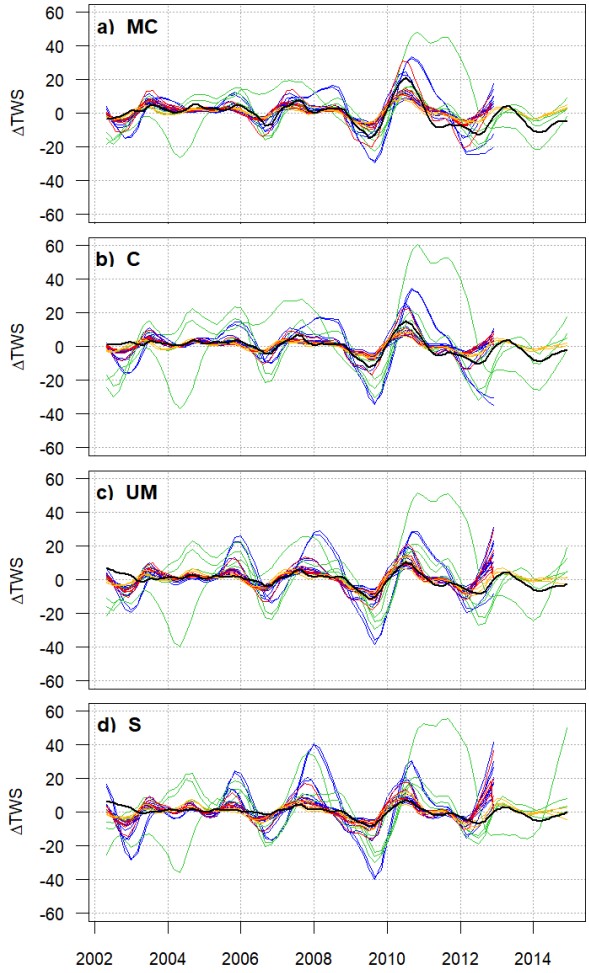

**Figure 10.** Long-term trends time series for the models and GRACE for a) MC, b) Cauca, c) UM, and d) Saldaña basins. The black line indicates the GRACE JPL, the blue and green lines the GHM WRR1 and WRR2 respectively, the red lines the LSM WRR1 and the yellow lines the LSM WRR2.

extensively regulated basin, though reservoirs in the Limpopo primarily serve irrigation and flood control, the operation of which may be better captured by the generic reservoir operation rules. This underlines the challenge of simulating reservoirs
and their sector-specific operation in (global) hydrological models (Rougé et al., 2021).

Figure 11 shows that over the MC basin as a whole, the TWS change computed from W3RA WRR2 using Eq2 best represents the signal of the GRACE data; including the monthly series, seasonality and long-term trend. The performance metrics do demonstrate, however, that for the different sub-basins, the best performing models include both GHM and LSM. Although the W3RA model is among the best performing models in all basins, for the monthly series JULES R2Eq1 performs best in





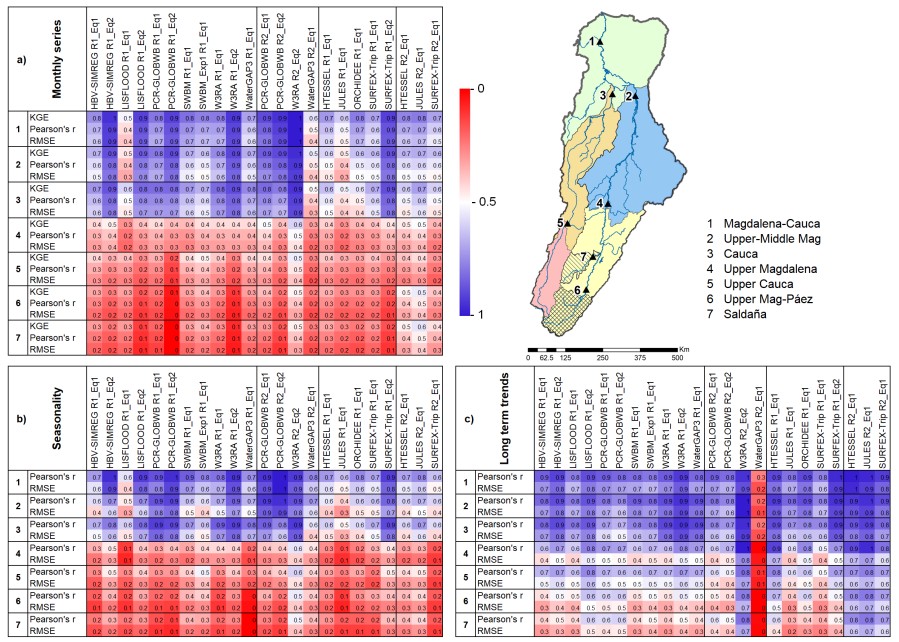

**Figure 11.** Summary figure of the models performance respect to GRACE data. Statisticians of KGE, Pearson's r, and RMSE were rescaled between 0 and 1, with 0 being the worst performance among the models and 1 being the best performance. The evaluation for the monthly time series is presented in a), in b) the evaluation for the seasonality, and in c) for the long-term trends.

the smaller basins. Besides W3RA, the seasonality is equally well captured by PCR-GLOBWB R2Eq2, while for long-term trends, HTESSEL R2Eq1, and JULES R2Eq1 (both LSM) also show good agreement to the GRACE data. Each of these models performed well with similar statistics throughout the basin, despite the differences in structure.

     Zhang et al. (2017) evaluate the TWS estimates from four hydrological models (two GHM and two LSM) and GRACE in 31 of the world's largest river basins. Although the MC basin is not included in their study, they similarly find that the performance

of the models varies from basin to basin, even within the same climate zone. They conclude that the variation in performance could be due to model structure, parameterization, the different water storage components included in the calculation as TWS, as well as being connected to the differences in runoff simulating and evaporation scheme (Zhang et al., 2017; Ramillien et al., 2006). In our case, the best performing models, W3RA, HTESSEL and JULES calculate evapotranspiration using the Penman–Monteith method, and saturation and infiltration excess for calculating runoff. However, the worst performing models,

LISFLOOD and SURFEX-Trip use these same schemes, while WaterGAP3 uses the Priestley–Taylor method for evapotranspiration and Beta function for runoff. The routing scheme, whether human water use is incorporated, and time step can also cause significant differences between the models. Within the ensemble of models considered here there are, however, multiple differences between models structures, making it difficult to uniquely identify model structures that singularly contribute to


improved performance. This would imply that the relationship between model performance and model structures that best
represents the processes in a tropical basin such as the MC is as yet inconclusive.

However, Schellekens et al. (2017) note that the W3RA model applies a Gash event-based model to simulate interception,
an important hydrological process in tropical basins (Miralles et al., 2010), and this could contribute to its relatively good
performance. Most models tested here follow a single reservoir and potential evaporation process to compute interception,
while other models do not include an interception scheme. Additionally, models that include a groundwater component and for
which the the performance can thus be evaluated using Equation 2 show better performance over the same models when using
Equation 1 to compute TWS (Fig. 11). Notwithstanding, further analysis is required in which the representation of internal
process, evapotranspiration, and runoff approaches is evaluated in-depth. It is also important to note that the ranking of model
performance may be quite different in different basins, as well as depending on the aim of the research. However, we underline
the importance of using a fundamental variable such as TWS as a proxy to inquire internal model states to benchmark model
performance.

## 4.2    Performance in representing seasonality and long-term trends

The differences between models in representing TWS change seasonality and long-term trends may be due to different reasons.
Following Scanlon et al. (2019), differences in seasonal amplitudes of TWS between models and GRACE can result from
uncertainties in models, in GRACE, or in both. These uncertainties may come from the scheme of modelled storage capacity
and storage compartments included in each model (Table 2), uncertainties in modelled inflows/outflows, and uncertainties
in modelled human interventions in the case of GHM, or lack of these in LSM. Storage capacity and compartments such
SWS and GWS are critical in tropical basins, where the magnitudes of TWS seasonal amplitudes are high, driven by seasonal
precipitation (Scanlon et al., 2019). The average bi-modal behaviour is evident in the seasonal signal of TWS from GRACE
and models, but the peak in the SON season in the models is greater in the smaller basins. The peak in the SON season
logically follows the precipitation used to force the models, but in GRACE this peak is generally lower than the MAM peak.
The overestimation (underestimation) of modelled seasonality of TWS change relative to GRACE in the study area could result
from underestimation (overestimation) of the storage capacity. A possible explanation of the differences in the peaks observed
in these two wet seasons as observed in GRACE is that the MAM wet season is preceded by a strong dry season. As the soil
receives a large amount of water, this infiltrates and recharges groundwater and thus adds to the TWS in the basin; the soil
becomes saturated; and the remainder becomes surface runoff. For the second wet season (SON), which is generally stronger
and with greater precipitation quantity (Guzmán et al., 2014; Poveda, 2004), the soil is not as dry as in the first wet season, and
therefore soils saturate more rapidly generating more runoff that quickly leave the basins, with smaller change in TWS (e.g. in
groundwater) as consequence.

Due to the poor representation of the storage capacity in the models, these may be overestimating changes of TWS in
SON especially in the South, where the topography is more complex. The heterogeneity of climate in the basin may also be
important as there is a tendency to a unimodal climate towards the North of the basin with a much less pronounced JJA dry





season (Urrea et al., 2019). Pokhrel et al. (2013) establish similar conclusions in the Amazon basin where model representation of the groundwater - vadose zone - surface water dynamics is important in representing seasonal dynamics.

Similarly, discrepancies in long-term trends in TWS may be related to uncertainties in models and/or GRACE. Scanlon et al.
(2018) evaluate seven different global models against GRACE. Considering initial conditions, water storage compartments and capacity related to model structure, precipitation uncertainty, and model calibration, they conclude that the models considered underestimate large decadal water storage trends relative to GRACE data. In this study, the models have the same input/forcing for each WRR and are not calibrated, so differences must be due to model structure (e.g. representation of water storage compartments) and parameterization (e.g. capacity of compartments) (Dutra et al., 2017). The way each model computes
the storage capacity could be related to the lack of storage compartments, soil profiles (thickness and number of layers), or exclusion of processes such as river flooding (Scanlon et al., 2018). One of the clear factors is that most LSM do not model SWS and GWS compartments, with the exception of Surfex-Trip. However, the inclusion of SWS and GWS is not conclusive for a good agreement with GRACE in our study as Surfex-Trip overestimates long-term trends and the LSM in general have a better performance than most GHM (Fig. 7c,d and Fig. 11c). Models also differ in how storage compartments such as the soil
layer are discretized (Schellekens et al., 2017). While W3RA has three soil layers and shows good agreement, WaterGAP has only one, and SurfexTrip has fourteen soil layers. More than the number of soil layers, the thickness and total soil depth could be an important variable for storage capacity calculation. Swenson and Lawrence (2015) report that the thickness of the profile required to replicate GRACE TWS variability is up to 8–10 m in tropical regions (e.g., Amazon, Congo) and in South Africa, while most model considered here have a soil thickness of between 1 m and 4 m (Schellekens et al., 2017; Dutra et al., 2017),
thus limiting the storage dynamics.

## 4.3 Basin scale analysis

As basin size reduces, the performance of models is found to decrease. Here it is important to highlight that GRACE measurements and leakage uncertainties increase with decreasing basin size (Scanlon et al., 2016). In Figures 4 and 7, the shift occurs between the Cauca ($\sim$ 60,657 km$^2$) and UM ($\sim$ 56,992 km$^2$). Currently, basins with a size of $\sim$ 63,000 km$^2$ can be resolved
to an error level of 2 cm in terms of equivalent water height (Vishwakarma et al., 2018), which would agree to the size smaller than which we find a marked decrease of performance. Notwithstanding, when we analyse the spatial correlation maps (Figs. 5 and 6) and the seasonal maps (Fig. 9), we observe that there are large discrepancies for the South of the macro-basin.

The Southern part of the basin corresponds to an area with a more complex topography. While the global models in general provide a poor representation of the wetlands in the North of the MC basin, the hydro-climatology of the mountainous areas is
more heterogenous as precipitation not only depends on the macroclimatic phenomena such as the ENSO and ITCZ migration, but is also influenced by other atmospheric circulation mechanisms such as meso-scale convective Systems, soil-atmosphere interaction processes, and local circulation patterns (Poveda, 2004). Also, it is important to highlight the presence of high-altitude montane wetlands (Paramos). These are one of the most important ecosystems in Colombia and provide an important source of water supply for many of the big cities in the Andean region (Rodríguez and Armenteras, 2005). The hydrological
processes in these Paramos are poorly understood, making the simulation of the water balance and storage dynamics diffi-



cult (Buytaert and Beven, 2011). In addition, the upper basins are highly intervened by human activities, including several reservoirs, which challenges the models in simulating flows and water storage compartments. Gründemann et al. (2018) point out that models that capture only natural flow conditions, and do not take artificial reservoirs and water usage into account, may be able to reasonably estimate runoff volumes, though they do tend to overestimate the actual magnitude of discharges.

Their results suggest that GHM would have better performance over LSM, in particular those that include human interventions. While we do observe this to be the case for the general MC as well as selected sub-basins (UMM and Cauca), when comparing monthly time scale of TWS for the smaller basins, we find that the LSM WRR2 have a better agreement over GHM (Fig. 11).

Figure 11 also shows that the deterioration of model performance for basins smaller than $60,000 \text{ km}^2$ is most clearly seen for those models where GRACE TWS is used to benchmark models that represent internal model stocks, such as groundwater,
using Equation 2. Benchmarking those same models using Equation 1 results in a much more varied picture, which is also seen in the results of Rodríguez et al. (2019) who tested several of the global models used here as well as (calibrated) regional models against observed discharges from 88 gauges across the MC basin. This did not reveal a clear patterns in model performance across the basin.

## 5 Conclusions

With the overall poor and decreasing availability of observed hydrological data in tropical basins, there is an increasing interest in the use of remote sensing data and models to study water resources, as well as the impact of climate change and human influences on water resources. The recently developed Earth2Observe (E2O) global water-resources reanalysis dataset provides hydro-meteorological data of sufficient length and coverage to complement observed data in water resources assessments. However, insufficient availability and poor spatial distribution of observed data mean independent evaluation of these reanal-
ysis data in representing key hydrological processes and storage dynamics in these basins is difficult. The GRACE satellite dataset on the other hand, is a recent and powerful tool that provides independent and distributed observations of Total Water Storage (TWS) in river basins, giving insight into the water storage dynamics that would not be possible through conventional observations. These data provide an opportunity to benchmark water storage dynamics of the models used in water-resources reanalyses and explore the question of how well these represent basin TWS dynamics.

We evaluate the representation of TWS change from ten models in the E2O dataset, including six Global Hydrological Models (GHM) and four Land-Surface Models (LSM). We benchmark the performance of these models in the Magdalena-Cauca basin in Colombia, a medium-size tropical basin that can be considered unique to tropical basins, given its comparatively well-developed monitoring network. We assess the potential of the two global Water Resources Reanalysis (WRR1 and WRR2) available from the E2O dataset by studying the performance of these models in simulating TWS change through commonly
used statistics.

Performance statistics reveal that the variability of GRACE TWS is better captured by the models in the larger basins compared to performance in the smaller, nested, basins. Basin sizes at which both WRR1 and WRR2 models are able to provide a better representation of the hydrological behavior are observed for areas above around 60,000 km$^2$, with significantly poorer



performance for smaller catchment sizes. Although this could be attributed to the relatively coarse resolution of the GRACE TWS data used as a benchmark and consequent uncertainty and signal leakage in smaller basins, we observe that the inadequate ability of the models to capture the hydrological process and storage dynamics in the South of the basin contribute to this poor performance. This part of the basin has a more complex topography and a higher degree of human intervention, such as reservoirs. Additionally, there are high montane wetlands (páramos) that have an essential role in regulating water resources. Although our benchmark does not reveal specific model structures that contribute to improved performance of either GHM or LHM, our results do suggest that the representation of processes such as interception and storage dynamics in the soil column, including groundwater, contributes to better model performance.

In general, models included in the higher resolution WRR2 dataset have a better performance than the models in the lower resolution WRR1 dataset. This shows that the continued efforts to improve global models, either through improved and higher-resolution forcing or improved and higher-resolution model structures and parameterizations, can enhance the models' ability to reproduce observed TWS and simulate water resources variability. However, poor representation of specific processes such as reservoir operation through a too generic scheme common to global models may obscure these improvements.

Our comparison highlights the relevance of using independent, remote sensing data to benchmark large scale models in specific hydro-climatological settings, including tropical basins. Our results suggest future directions for model development, highlighting the appropriate representation of water stocks and related processes. It is relevant that models that do include explicit representation of the internal storage dynamics allow for a more direct benchmarking of modelled TWS against the TWS data from GRACE. This would improve the assessment of model performance compared to benchmarking models against indirectly calculated TWS variability as derived from the balance of precipitation, evaporation, and observed basin outflow; thus helping discriminate models with better structures and process representation.

Although the disparity between GRACE and the models is subject to uncertainties in both, the quality of GRACE data will be continue to improve in the foreseeable future. In this study, we use the most recent GRACE release (RL5) before the constellation of satellites that provided these data discontinued their service. A GRACE follow-on mission as a successor to the original GRACE mission was launched in May 2018, with data shortly becoming available for future research. These continued advances in GRACE data and its use to independently benchmark internal states of hydrological models will improve our understanding of water resources and reduce uncertainties in using these models in water resources projections under climatic change and increasing human interventions.

*Data availability.* Data used in this manuscript are freely available in the following web pages: JPL mascon data were retrieved from the Tellus website https://grace.jpl.nasa.gov/data/get-data/jpl_global_mascons/ (Watkins et al., 2015); Earth2Observe models data were downloaded from the E2O Water Cycle Integrator portal https://wci.earth2observe.eu/ (last access: 20 November 2018). Additionally, we present the TWS data from GRACE and the estimated values for each model derived from the E2O data for the study area in the Supplementary Material.



*Author contributions.* SB and MW contributed to the conceptualization of this study. SB carried out the formal analysis, investigation, and prepared the original draft. MW and JFS reviewed and edited the final draft. All the authors reviewed early manuscript drafts and the final draft.

*Competing interests.* The authors declare that they have competing interests as follows: Micha Werner is editor of HESS.

*Acknowledgements.* This research was carried out with the financial support from the Ministry of Science, Technology, and Innovation of Colombia - MINCIENCIAS (previously COLCIENCIAS) through the National Doctorates Program N° 727/2015. We kindly acknowledge the Universidad de Antioquia and the project Evidence4Policy (project reference 106471), as part of the IHE Delft Partnership Programme for Water and Development (DUPC2), for supporting this study.



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
