# Peer review of "Benchmarking global hydrological and land surface models against GRACE in a medium-size tropical basin"

_Hydrology and Earth System Sciences, 2021_

## Author Comment (AC1)

**Response to Reviewer No #1**

We would like to thank the anonymous reviewer #1 for the detailed analysis of the manuscript and constructive comments. In this document we reflect on the comments made by the reviewer and how we propose to change/improve the manuscript in response to the issues raised. These comments are included for convenience (in blue font) as well as a detailed response to the comment and changes proposed to the manuscript.

In this study the authors use GRACE JPL mascon data to evaluate simulated total water storage (TWS) for 10 land surface (LSM) and global hydrological models (GHMs) over the Magdalena-Cauca basin (Colombia) and its sub-basins. They find different abilities of the different models to represent trends, seasonality and monthly time series, with model accuracy reducing from trends/seasonality to time series, from higher to lower resolution models and from larger to smaller basins. One of the models is declared the overall winner of the comparison.
I have the following comments:

Although this is an interesting and worthwhile exercise in itself, I am a bit hesitant about the novelty of this study. What exactly are the general conclusions we can draw from applying specific models to a specific basin? Global comparisons have been made before, as also testified by the references in the paper (Scanlon et al., 2016;2018; Schellekens et al., 2017). What does a regional study add to that? Does a study like this fit a general purpose hydrological journal like HESS, or does better fit a more applied journal that publishes well executed case studies? I leave it up to the editor, but if it is accepted, the authors should make clear what is novel about this work.

We thank for reviewer for noting that the study is of interest and worthwhile. We appreciate the point raised as to the novelty of the study, and agree that there have been several global comparisons (as referenced in the comment), as well as comparisons for individual basins (references provided in the paper). However, our study adds a novel contribution to these previous studies in two key ways. First, through developing a more detailed comparison at the basin level, which includes nested basins, allows a more comprehensive consideration of the importance of heterogeneous hydrological processes across the basin on the performance of these models against the benchmark provided by GRACE. This includes the poor representation of the dynamics of the wetlands in the lower basin, as well as the dynamics of TWS in the complex terrain of the upper basin. A second dimension that this study contributes is that the Magdalena-Cauca basin is unique amongst more detailed basin studies of GRACE as it is a tropical basin with a dominant monsoonal climate and pronounced ENSO influence (in parts of the basin), that also has a reasonably extensive and publicly available observed hydrometeorological dataset. This allows insight into model structures in LHM and GHM that would appear to provide a better representation of hydrological processes relevant to tropical basins to be obtained, and thus contribute to improving these also for basins in other tropical areas of the world that are not as well endowed with observational data.

To emphasize the contribution of the manuscript we propose the following changes:

Lines 9-10: a medium sized tropical basin with a well-developed gauging network when compared to other basins at similar latitudes.

Line 20-24: We conclude that GRACE provides a valuable dataset to benchmark global simulations of TWS change, in particular for those models with explicit representation of the internal dynamics of hydrological stocks, offering useful information for the continued models improvement in the representation of the hydrological dynamics in tropical basins around the globe.

Line 110-118: With the purpose to contribute to the understanding of the dynamic nature of TWS as well as to contribute to future LSM and GHM development and improvement, this study highlights the value of using water storage from GRACE, in addition to traditional water fluxes, as a benchmark in assessing global models in a tropical basin. The Magdalena-Cauca basin offers a special opportunity to compare global models for tropical

basins, it has a dominant monsoon climate and a pronounced ENSO influence in some parts of the basin, and it also has a reasonably extensive and publicly available observed hydrometeorological data set. On the basis of Magdalena-Cauca tropical basin is unique amongst more detailed basin studies of GRACE, this assess allows insight into model structures in LHM and GHM, and contributes to improving these also for basins in other tropical areas of the world that are not as well endowed with observational data. Assessing models using these recently available data of an important state variable such as TWS, can contribute to a better understanding of the hydrological cycle processes, with the improvement in the modeling and forecasting of hydrological variables in tropical basins, thus being conductive to better tools for decision-making around water management and sustainability. The relatively large set of LSM and GHM models considered in this study are obtained through the open access global Water Resources Reanalysis dataset developed in the eartH2Observe (E2O) research project, a collaborative project funded under the European Union's Seventh Framework Programme (EU–FP7) (Schellekens et al., 2017).

Using GRACE that for sub-basins below 40000 km2 in size is very tricky, even if mascons are used. The inherent resolution of GRACE is too coarse for this. This means that the results for the smaller basins are questionable at best, and the differences between GRACE partly from the models and partly from the GRACE estimates. The question then is which part of the deviation comes from the models and which part from GRACE. The authors should either leave out the smaller basins or be very upfront about this limitation in the Introduction/Methods section already and not wait until the Discussion.

We agree with the reviewer that using GRACE for sub-basins below 40000 km2 is very tricky and indeed raise this in the discussion. In our opinion, leaving the smaller basins out of the analysis would quite substantially change the contribution of the manuscript. As such we follow the reviewer's suggestion to be quite up-front about this.

We propose to add a sentence in the introduction in lines 83-88:

Line 83-88:
Although GRACE has important limitations due to its resolution (Chen et al., 2016), data from GRACE do provide a uniquely independent estimate of the distributed TWS in a river basin as water balance estimation based on observed data and models require gauging data (which are often deficient or insufficient) or data from reanalysis models, which are not direct observations. Advances in GRACE processing from traditional spherical harmonics to more recent mass concentration (mascon) solutions have increased the signal-to-noise ratio and reduced uncertainties (Scanlon et al., 2016), though the interpretation of results for basins smaller than on the order of ~40,000 km2 remains difficult due to the inherent coarse resolution of GRACE data (Scanlon et, al. 2016; Vishwakarma et al., 2018).

Line 33-35: The argument that knowing TWS leads to better forecasts is often used. Please provide us with examples from the literature where it is shown that significantly better streamflow forecasts are obtained when GRACE TWS is ingested into the model?

Our comment here is that a better representation of the basin initial state contributes to improved (streamflow) forecasts, in particular in basins where there is significant internal storage and persistence of initial states. Examples that have shown the specific contribution of GRACE TWS in improving streamflow forecasts include a recently published study by Liu at el, 2021 (https://doi.org/10.1080/02626667.2021.1998510), as well as Getirana et al 2020 (https://doi.org/10.1175/JHM-D-19-0096.1). These references will be included.

Line 35-42: I have to say that this argumentation is a bit silly. Before GRACE, nobody cared about the validating TWS of hydrological models at all! The reason is that it could not be observed. Before GRACE, only partial state variables, such as groundwater, river and lake levels, soil moisture and SWE were independently evaluated using in-situ and remotely sensed data. Only after GRACE, TWS anomalies could be validated and were therefore

computed from models.

We appreciate that before the availability of GRACE the interest in TWS as a variable may not have been apparent as it could not be observed. However, the point we had intended to raise is the importance of monitoring or estimation of change in water stocks as represented by TWS to water resources assessment, and that given the difficulty of independent integrated observation, water resources assessments are commonly developed using models and water balances.

Despite the acknowledged importance of this variable, prior to the availability of data from GRACE integrated observations of water stocks at the basin scale, as represented by TWS, were unavailable, with only partial state variables such as groundwater levels, soil moisture, river and lake levels, and snow water equivalent available from direct in-situ and remotely sensed observations. Given the heterogeneity of the hydrology of river basins, comprehensive observation of these is, however, very difficult due to insufficient in-situ observations of these partial variables, further confounded by the global decline in gauging networks (Hassan and Jin, 2016). Estimation of TWS and its change at the basin scale is therefore commonly done through water balances and the use of models. Given the difficulty to measure TWS (Tang et al., 2010) these. Many traditional analyses assume that at longer timescales and over large regions, change in TWS can be approximated as zero. This implies that in water balance studies it is common to ignore the long-term trends of TWS (Reager and Famiglietti, 2013).

Figure 11 shows that WaterGap and Lisflood both show relatively poor performance in reproducing TWS anomalies. What is striking is that these models both have been subject to some sort of calibration to streamflow data (see the paper by Beck et al., 2017 where they perform very well in streamflow reproduction). Could it be that calibrating GHMs to streamflow only (without constraining internal states and fluxes by other information) has led to correcting errors in streamflow by accruing errors elsewhere in the model?

We thank the reviewer for underlining the poor performance of these two models, in particular, for the R1 dataset. The performance of WaterGap in the R2 dataset is also quite clear, and is discussed extensively in the discussion (note that LISFLOOD was not available using in the R2 dataset). We agree that the introduction of calibration of models against observed discharges may improve model results at the basin outlets, which may be detrimental to representation of internal states and fluxes. This may be particularly so where there are biases in the forcing data over the complex topography of the basin (see also response to Reviewer #2). The paper of Beck will be added to improve the discussion.

**References**

Beck, H. E., van Dijk, A. I. J. M., de Roo, A., Dutra, E., Fink, G., Orth, R., and Schellekens, J.: Global evaluation of runoff from 10 state-of-the-art hydrological models, Hydrol. Earth Syst. Sci., 21, 2881–2903, 2017.

Scanlon, B., Zhang, Z., Save, H., Sun, A., Schmied, H., van 630 Beek, L., Wiese, D., Wada, Y., Long, D., Reedy, R. C., et al.: Global models underestimate large decadal declining and rising water storage trends relative to GRACE satellite data, Proceedings of the National Academy of Sciences, 115, E1080–E1089, 2018.

Scanlon, B., Zhang, Z., Rateb, A., Sun, A.,Wiese, D., Save, H., Beaudoing, H., Lo, M., Müller-Schmied, H., Döll, P., et al.: Tracking seasonal fluctuations in land water storage using global models and GRACE satellites, Geophysical Research Letters, 46, 5254–5264, 2019.

Schellekens, J., Dutra, E., la Torre, A. M.-d., Balsamo, G., van Dijk, A., Weiland, F. S., Minvielle, M., Calvet, J.-C., Decharme, B., Eisner, S., et al.: A global water resources ensemble of hydrological models: the eartH2Observe Tier-1 dataset, Earth System Science Data, 9, 389–413, 2017.

---

## Author Comment (AC2)

**Response to Reviewer No #2**

We would like to thank the anonymous reviewer #2 for the detailed analysis of the manuscript and constructive comments. In this document we reflect on the comments made by the reviewer and how we propose to change/improve the manuscript in response to the issues raised. These comments are included for convenience (in blue font) as well as a detailed response to the comment and changes proposed to the manuscript.

This manuscript highlights the use of an external source of data (JPL GRACE TWS monthly anomalies) to benchmark 10 different global hydrological and land surface models using results of the earth2observe project, in a well-instrumented tropical basin in Colombia, the Magdalena-Cauca (MC) macrobasin as the area of study. Findings identify characteristics and limitations of the models and are a key input for contributing to identifying new developments and improvements of these types of models.

The article is well written, organized, and discusses nicely the main findings. The objectives the paper sets out to are of interest, and there is scientific merit for publishing it. Below are some specific comments to the authors:

We thank the reviewer for the positive comments on the scientific merit and structure of the article.

In the abstract (line 11) and the methodology, analysis and long-term tendencies in terrestrial water storage (TWS) are based on JPL GRACE data from 2002-2014. What are the limitations of these estimations taking into consideration that the period is short (only 13 years), that the MC has a large inter-annual climate variability associated with the ENSO and other phenomena, and that the base period used to calculate the anomalies is also short (2004-2009)?

We thank the reviewer for the comment, and agree that the length of the data is not that long, but by long term tendencies we actually mean variability at the ENSO time scales (i.e. multi-annual to decadal time scales) and not longer than that (e.g. climate time scales). According to the article Bolaños et al, 2020, in the long-term analysis, the effect of climate change on TWS is not conclusive due to the short time series. Therefore, the observed trends may be mainly due to the climatic variability present in the region.

This study focuses more specifically on long-term variability. In the study, the long-term series is obtained through Seasonal Trend decomposition by Loess, (STL) proposed by Cleveland et al. (1990) to estimate the relative magnitudes of water storage variance of different time series components. This is done in order to compare the seasonality and long-term trend of the models with respect to GRACE, and to observe how they capture the climatic variability in the tropical region. With the new data collected by GRACE Follow-On, it will be possible in the future to have a longer TWS series that allows better long-term analysis.

Although it is not completely clear in the manuscript, because it is not explicitly mentioned in the Data and Methods section, it seems (see line 103, line 221) that TWS is calculated from the models´ results and the JPL GRACE data at the macrobasin and subbasins scale using the average of the values for all the cells in the corresponding domain and time step. If this is true, this approach could have some limitations that the authors should address within the discussion and conclusions. And if not, an explanation of the methodology used and its limitations should be included in the manuscript.

In fact, the TWS anomaly data was averaged within the time step and domain.
So, we add the follow sentence in Line 196 to clarify the method: For both TWS calculated from models and JPL GRACE data, the values of all cells were averaged corresponding to each time step to construct the time series for each database and each watershed. This implies some limitations such as sensitivity to extreme values, and the averaging of biases, therefore, in this study we assume that the variables are normally distributed.

In lines 170 and 418 it is important to consider that from WRR1 to WRR2 some models also have some type of calibration, not necessarily in the MC basin.

We thank the reviewer for the comment, and agree that it is important to mention that indeed some of the models in both WRR1 and WRR2 are calibrated. Models that have been calibrated include LISFLOOD, WaterGAP, HBV-SIMREG and SWBM (see also comments by reviewer #1 and Beck et al., 2017b).

We add the following sentence in line 170:
Selected models in WRR2 and WRR2 have been calibrated against streamflow data, including LISLFOOD, WaterGAP, HBV-SIMREG and SWBM (Beck et al., 2017b).

We also include an indication of the models that have been calibrated in Table 1.

The sentence in line 418 has been amended:
In this study, the models have the same input/forcing for each WRR and the majority of models have not been calibrated, while for those models that have been calibrated (Table 1) these calibrations were not specific to the MC basin (Beck et al., 2017b). This implies that differences must largely be due to model structure (e.g. representation of water storage compartments) and parameterization (e.g. capacity of compartments) (Dutra et al., 2017).

The legend used for the different models and modelling phases (WRR1 and WRR2) is consistent throughout the document. However, the first time the legend is introduced is in Table 2. Perhaps an explanation of the legend in this Table would facilitate the analysis right from the beginning of the paper.

We have included the legend by which models are identified in Table 2 (e.g. R2eq1) and have also amended the title of the table to clarify.

**Table 2.** Components used in TWS change estimation for each model. The last four columns included the symbols and legends used to identify model resolution and equation used to derive simulated TWS change. These are used throughout the manuscript.

Equation 3 proposes a way to decompose the time series of TWS into seasonality, long term, and residuals. For the first two components, a detailed analysis is conducted. However, for the residuals, it is not the case. The analysis of the residuals would be a nice way to complement the findings of the study.

In line 335 perhaps the analysis of the residuals quite nicely complements the results.

Equation 3 proposes a way to decompose the time series into seasonality, long-term, and residuals through the Seasonal Trend Loess (STL) decomposition. However, the main objective of the study was to compare the seasonality and long-term variability of the models with GRACE.

Much of the time series variability that we can observe is included in seasonality. Furthermore, an analysis of residual correlations was done, and the results were very poor, so we decided not to include it in the study.

In Figure 2 they appear 7 different GHM including SWBM_Exp 1 (in addition to SWBM). This experiment with this model is not described either in Table 2 or in the text. For consistency in the document, where 10 models are analyzed, this experiment should be dropped from the analysis.

In Figure 11 it also appears the SWBM_Exp1 model, which either should be described in the

manuscript considering 11 instead of 10 models or dropped from the analysis.

Thank you for this observation. We accept the suggestion.

In previous studies that have used discharge to investigate the performance of the models in the earth2observe project in the MC basin it has been shown that LISFLOOD obtains the lower results as it is also confirmed in this study (line 239). Reasons for the low performance of this model in the MC are not discussed in the document and would be helpful to include.

Thank you for this observation. In fact, LISFLOOD obtains lower results, and one reason could be that LISFLOOD is reported to underestimate quick flow response, which could lead to less pronounced seasonality (as shown in Figure 9 ). PCR-GLOBWB is also reported to have this issue (see Beck, 2017b). On the other hand, this poor behavior could also be due to the calibration.
The discussion of this will be included in the document as de reviewer suggests.

In several parts of the article a threshold of 60,000 km2 has been proposed as the basin size limit for the use of GRACE data to validate the models. In this sense would be the Cauca (C) basin an exception? How do the different climatological regimes in the C and Upper Magdalena (UM) basins influence the results? It is evident that for the small basins including UM, Upper Magdalena Paez (UMP), and Saldaña (S) results are poor and this is the reason for choosing the size limit proposed. However, right from the start results in the UM are poor, so for other subbasins in this area, it would be expected that results are also poor. What would happen if instead of considering subbasins in the UM you choose subbasins in the C (additional to the Upper Cauca (UC), where the size is small and surely below the limit), where results are much better?

In the study, only five subbasins have drainage areas close to or below 60,000 km2. Considering the climatological and physical complexity of the MC macrobasin, in my opinion, there is not enough information to establish the threshold proposed as a basin size limit for evaluating model performance against GRACE data.

In this study, we are not suggesting a threshold as such, but we do observe a marked difference for the basins above/below this basin size. In the bibliography, Vishwakarma et al., 2018 propose a basin size limit of ~63,000 km2, which is close and consistent with what was found in this study. On the other hand, although in UM, UMP, and S the behavior is poor (such as being expected since UM subbasin is poor), we consider them in the study because they comprise the upper area of the basin, which has complex topographic characteristics and the presence of moors, which models fail to represent adequately.

Following the previous comments, for the UM and C basins, with approximately the same size, there is quite a contrast in the results. For the first subbasin, results are way lower than for the second one. Similar results in the UM that the ones presented in the study have been obtained with several different models, not only global but also regional and local. In this sense, any model structure seems to perform poorly in the UM. Problems in the precipitation forcing used for this basin could be part of the reason? Recent studies (unpublished) have shown that in some basins of the UM, including the S, the monthly precipitation and discharge average patterns do not match. Rainfall is mainly bimodal, as captured by the models´ forcings in this study, but streamflow is mainly unimodal. This could be associated with anthropogenic interventions, clearly discussed in the manuscript, but also with climatological forcing limitations that need to be addressed in the paper.

In line 448 besides the reasons for the poor performance of the models in the UM, perhaps influence from the Orinoco and Amazon macrobasins, may also play a role in the results. Some consideration about this is also recommended to be included in the discussion.

Thank you for this observation. We included this in the document as reviewer suggests.

We add the following sentence between line 441 and 442.

Line 441: … but is also influenced by other atmospheric circulation mechanisms such as meso-scale convective Systems, soil-atmosphere interaction processes, and local circulation patterns (Poveda, 2004). Furthermore, the interplay between the Orinoco and Amazon basins plays an important role in the moisture availability for UM precipitation, which would add complexity to the hydro climatological characteristics in this region, and a huge challenge for models and their calibration.
It is important to highlight in the UM basin, the presence of high altitude montane wetlands (Paramos)…

In line 274 it should be Figure 4c instead of Figure 5c

This has been changed as suggested.

Figures 5 and 6 (line 282, line 309) in my opinion could be included in the supplementary material, as they are not key for supporting the main findings described in the article. Instead, the analysis of the residuals perhaps could better support the analyses and discussion.

We agree with the reviewer and have moved these to the supplementary material as well as the references to these figures.

In line 283 it should be In these figures… instead of In this Figure ….

This has been changed as suggested.

For Figure 4 there is enough space in the graph to include the accompanying legend to facilitate the interpretation of the results.

We have added the legend to Figure 4.

Sentence in line 321 is not clear.

The sentence has been amended as follows:
… which means that these dry out too much in the drier DJF period and cannot represent the increased storage in the wet period in SON in La Mojana area, which can be observed by GRACE.

Results for the WATERGAP3 model in the Limpopo River Basin have shown the good performance of this model (line 356). Results in the MC and some of its subbasins have also shown good results for this model when discharge observations are used. How to interpret that when using GRACE data as a complementary source of validation, results for this model deteriorate so much?

We infer that this could be due to the calibration and representation of internal model states. The model could adequately represent the discharge but the internal hydrological processes could not be right to underestimate/overestimate some fluxes, which means that the discharges are reasonable, but that this is to the detriment of internal model states. Since TWS is a state variable, it gives an approximation of the internal hydrological processes that result in discharges in the basins. Therefore, including TWS in the models can improve the representation of flows without detriment to their internal states.

Instrumentation in the UM, especially in the higher altitudes could in my opinion help to separate the influence of the anthropogenic interventions from the limitations in precipitation forcing and how they impact the streamflow patterns observed for this part of the MC catchment.

We agree with the reviewer y we thank you for this comment. In the document, we discuss that UM poor performance could be due to different reasons or the sum of all limitations in hydroclimatologic calibration, complex topography it this region, and influence of the anthropogenic interventions in the basin. On the other hand, the Cauca basin could be more strongly influenced by climate teleconnections which could contribute to its better performance in contrast with UM.

**References:**

Beck, H. E., van Dijk, A. I. J. M., de Roo, A., Dutra, E., Fink, G., Orth, R., and Schellekens, J.: Global evaluation of runoff from 10 state-of-the-art hydrological models, Hydrol. Earth Syst. Sci., 21, 2881–2903, 2017b.

Bolaños, S., Salazar, J. F., Betancur, T., and Werner, M.: GRACE reveals depletion of water storage in northwestern South America between ENSO extremes, Journal of Hydrology, p. 125687, 2020.

Cleveland, R. B., Cleveland, W. S., McRae, J. E., and Terpenning, I.: STL: a seasonal-trend decomposition, Journal of official statistics, 6, 3–73, 1990.

Vishwakarma, B. D., Devaraju, B., and Sneeuw, N.: What is the spatial resolution of GRACE satellite products for hydrology?, Remote Sensing, 10, 852, 2018.

---

## Author Response (AR3)

**Response to Editor**

Checking your paper, I noticed that your table 2 contains coloured cells. Please note that this will not be possible in the final revised version of the paper due to HTML conversion of the paper. When revising the final version, you can use footnotes or italic/bold font. For now, the process will continue, but please note that the final version cannot be published by using coloured tables.

Thank you for your comments. Table two has been modified without coloured cells.